# Analysis of Petrogenic Hydrocarbons in Plant Tissues: A Simple GC-MS-Based Protocol to Distinguish Biogenic Hydrocarbons from Diesel-Derived Compounds

**DOI:** 10.3390/plants13020298

**Published:** 2024-01-19

**Authors:** Elena Collina, Enrico Casati, Andrea Franzetti, Sarah Caronni, Rodolfo Gentili, Sandra Citterio

**Affiliations:** Department of Earth and Environmental Sciences, University of Milano-Bicocca, Piazza della Scienza 1, 20126 Milan, Italy; elena.collina@unimib.it (E.C.); enrico.casati@unimib.it (E.C.); andrea.franzetti@unimib.it (A.F.); rodolfo.gentili@unimib.it (R.G.); sandra.citterio@unimib.it (S.C.)

**Keywords:** diesel compounds, plant uptake, GC-MSD analysis, diesel phytoremediation, *Vicia sativa*, *Secale cereale*

## Abstract

Diesel contamination of farming soils is of great concern because hydrocarbons are toxic to all forms of life and can potentially enter the food web through crops or plants used for remediation. Data on plant ability to uptake, translocate and accumulate diesel-derived compounds are controversial not only due to the probable diverse attitude of plant species but also because of the lack of a reliable method with which to distinguish petrogenic from biogenic compounds in plant tissues. The purpose of this study was to set up a GC-MS-based protocol enabling the determination of diesel-derived hydrocarbons in plants grown in contaminated soil for assessing human and ecological risks, predicting phytoremediation effectiveness and biomass disposal. To this end, two plant species, *Vicia sativa* L. and *Secale cereale* L., belonging to two diverse vascular plant families, were used as plant models. They were grown in soil spiked with increasing concentrations of diesel fuel, and the produced biomass was used to set up the hydrocarbon extraction and GC-MSD analysis. The developed protocol was also applied to the analysis of *Typha latifolia* L. plants, belonging to a different botanical family and grown in a long-time and highly contaminated natural soil. Results showed the possibility of distinguishing diesel-derived compounds from biogenic hydrocarbons in most terrestrial vascular plants, just considering the total diesel compounds in the n-alkanes carbon range C10–C26, where the interference of biogenic compounds is negligible. Diesel hydrocarbons quantification in plant tissues was strongly correlated (0.92 < r^2^ < 0.99) to the concentration of diesel in spiked soils, suggesting a general ability of the considered plant species to adsorb and translocate relatively low amounts of diesel hydrocarbons and the reliability of the developed protocol.

## 1. Introduction

Diesel-contamination of soils is a widespread environmental problem due to the extensive use of diesel fuel as a source of energy and its associated disposal operations. Environmental contamination by diesel oil is also relatively common also because of illegal or petroleum collection from oil pipelines leading to accidental spills. This is a common practice in developing countries [1] and is becoming widespread in industrialized countries. In Italy, 465 illegal actions were detected between 2011 and 2019, with a peak of 165 attacks in 2015 [2,3].

In Europe, 45% of identified polluted sites are contaminated with petroleum derivatives [4]. Many of these sites are diesel-contaminated farming areas where the crops represent a pathway of exposition, potentially generating adverse effects to higher trophic level organisms. 

Diesel is a mixture of saturated hydrocarbons (60–80%, primarily paraffins) and aromatic hydrocarbons (20–40%, including naphthalenes and alkylbenzenes) that has been reported to be toxic to plants and humans [5]. For instance, among the diesel components, polycyclic aromatic hydrocarbons (PAH) are genotoxic to plants [6] and can cause carcinogenic, teratogenic and mutagenic effects in humans and animals subjected to exposure [7,8,9,10,11]. 

Given the potential risk of indirect exposure to agricultural soil contaminants for all forms of life, the Italian legislation for soil recovery (DL 152, 2006) was recently integrated with a specific section for areas dedicated to agriculture and husbandry (DM 2019, n.46). The new law considers the transfer and accumulation of contaminants in plant tissues to define the acceptable risk, as well as suggesting phytoremediation for the reclamation of this type of area, i.e., that which needs to maintain and restore soil functions for agricultural production. However, to date, the amount of hydrocarbon contaminants accumulated by different plant species during typical cultivation or phytoremediation, as well as their distribution in plant organs, is still controversial and difficult determine, including with respect to the assessment of whether plant uptake of diesel hydrocarbons presents a toxicological problem at higher trophic levels [12]. Indeed, in theory, the composition and the structure of the plant cuticle, cell wall and plasmodesmata allow for oil compound penetration and transport within the plant tissues via both apoplastic and symplastic routes [12]. In agreement, Naidoo and Naidoo [13] observed oil deposits within the root cap, meristem cortex, conducting tissues and mesophyll cells of mangroves. Also, Wild and coauthors [14] demonstrated the ability of maize and wheat to uptake petrogenic hydrocarbons; the authors observed the movement of anthracene and phenanthrene from the substrate to the cortex within the root passing through the epidermal cells. However, they observed that both the compounds were not translocated to the stem and leaves. In general, the literature studies indicate the capacity of plants to uptake hydrocarbons, although they provide variable results with respect to the extent of hydrocarbon accumulation in the different organs. Many factors contributed to the variability of the literature data such as the different experimental conditions and the intrinsic characteristics of the different plant species. However, the main factor explaining the existence of controversial data on the ability of plants to accumulate this type of organic substances is related to the presence of biogenic hydrocarbons in plant organs that make it difficult to detect and determine the concentration of such contaminants in plants. Plants synthetize many different compounds, such as n-alkanes, alkenes, sterols, fatty acids, waxes, etc., whose structures are hardly distinguishable from those of petrogenic hydrocarbons with current analytical methods [12]. Only a few studies have focused on the hydrocarbon petrogenic/biogenic origin in plants grown in agricultural soil contaminated by diesel oil [2]. Indeed, this is probably due to the assumption that hydrocarbons are not accumulated in plant tissues (CCME 2008) and/or to the lack of an analytical methodology specifically developed for the quantitative determination of petrogenic hydrocarbon content in plants. The current most-applied method (ISO 16703:2004) [15] was developed for soil analysis and is based on gas chromatography with a flame ionization detector (GC-FID). Most authors have used this methodology or other less-sensitive analytical techniques based on IR-spectroscopy, HPLC and gravimetry, all methods that cannot discriminate biogenic from petrogenic compounds [16,17,18,19]. This resulted in the impossibility of determining with certainty the quantity and type of hydrocarbons in plants cultivated on polluted lands. Moreover, the lack of a methodology suitable for plant analysis prevented the risk analysis and the biomass disposal from being carried out appropriately both during and at the end of the soil phytoremediation process.

The objective of this study was to set up a GC-MS-based protocol appropriate to disentangle the petrogenic content from the biogenic content in plants exposed to soil contaminated with diesel oil. Common vetch and and rye, two species belonging to different botanical families and widely used in the phytoremediation of hydrocarbons-contaminated soils, were used as model plants. In addition, common cattail plants, belonging to a different botanical family and grown in a long-time and highly contaminated natural soil, were analyzed to further validate the sensitivity and reliability of the developed protocol.

## 2. Materials and Methods

### 2.1. Experimental Design and Plant Growth Measurements

*Vicia sativa* L. (common vetch) and *Secale cereale* L. (rye) seeds were sown in pots filled with 3% organic soil contaminated with or without (control soil, S0) increasing concentrations of diesel fuel: 1000 mg kg^−1^ (soil S1); 5000 mg kg^−1^ (soil S2); 10,000 mg kg^−1^ (soil S3). These concentrations were chosen considering the tolerance of these two species to diesel compounds and the concentrations of diesel hydrocarbons usually present in the polluted sites. The aim was to obtain healthy plants with measurable contents of diesel-derived hydrocarbons inside their organs.

The 3%-organic-matter soil was prepared by mixing sterilized quartz sand (0.5 mm coarse grade) with autoclaved (20 min at 121 °C) sowing potting compost with the following characteristics: organic carbon (C) = 48%; organic nitrogen (N) = 1.5%; pH: 6.5. Three aliquots of soil were than spiked with the appropriate amount of commercial diesel fuel to obtain the above reported final concentrations. For each concentration, the contaminated soil was thoroughly mixed and then distributed into nine pots. A total of 36 pots (0.25 m diameter and 0.20 m deep), 9 filled with uncontaminated soil (S0) and 27 with contaminated soil, were prepared and allowed to stand for 2 weeks. For each experimental condition, three pots were used to germinate and cultivate common vetch and three for rye. The remaining three pots for each condition were used to check the role of plants and the natural attenuation in the expected reduction of soil contamination during the experiment (Table 1).

Before sowing, the total hydrocarbon content was quantified in three soil samples collected from each pot. Sixty-five seeds per pot were sown. Seeds of common vetch were inoculated with its specific rhizobial strains (AloscA^®^ group F, Padana Sementi Elette s.r.l., Tombolo, Italy) by mixing them with dry, clay-based granules containing rhizobia. Plants were grown under controlled conditions (25 °C; 12 h dark/12 h light, 150 µmol m^−2^ s^−1^) for 30 days and then harvested by upsetting the pots. The soil was carefully removed from the plants with a small brush and the roots were kept as intact as possible. Plants were than washed in distilled water and gently blotted with filter paper.

The experiment was terminated after 30 days as a longer duration under our experimental conditions would have affected the health of the plants.

Plant survival and plant growth were determined for each experimental condition. Plant growth was assessed by determining plant organ dry weight (DW) just after plant harvest. Leaves, stem and roots of each plant were separated, cut into small parts and placed in a dry cabinet at 40 °C until a constant weight was reached. Then, dry plant organs from each pot were weighed and underwent, along with the soil samples, the determination of hydrocarbon concentration.

Artificial soil was used instead of agricultural soil for growing common vetch and rye to facilitate the collection and cleaning of the roots.

*Typha latifolia* (common cattail) plants, grown in a soil from a highly diesel-contaminated site, were also used for the hydrocarbon determination. Specifically, diesel-polluted and unpolluted soil (control) were collected from the surface-ploughed layer of a paddy field located in the western part of Po plain (Italy). The area was characterized by a shallow water table, and the soils, mainly represented by Hydragric Anthrosols (Loamic; Ortieutric; Endoskeletic), had anthraquic conditions in the ploughed Ap horizon laying above an hydragric horizon [20]. This last horizon indicates that the water table level was floating inside the subsoil alternating period of anoxic and oxic conditions in addition to the usual surficial hydromorphic (gleyc) conditions due to rice cultivation. In this field, about 300 kg of soil was collected from a diesel-polluted subarea (about 18,000 mg kg^−1^) and 300 kg from an unpolluted subarea. Each of the two soils taken (contaminated and uncontaminated) was thoroughly mixed and then distributed into pots. Three pots, each containing about 100 kg of soil, were prepared for each condition (contaminated and uncontaminated). Three samples of soil from each pot were analyzed for hydrocarbon content before planting the cattail. Thirty 5 cm-long cat-tail rhizomes containing 3 active buds were simultaneously collected from the unpolluted subarea. Five rhizomes per pot were grown for 1 month and then harvested. For each plant, roots, rhizome and above-ground biomass were separated, cleaned, cut into small parts, dried, weighted and analyzed for diesel compounds.

### 2.2. Analytical Method for Diesel-Derived Hydrocarbon Quantification in Soil and Plant Material

Dried soils, shoots and roots were ground up to obtain fragments smaller than 2 mm. Dry soil (1.5 g) and plant samples (2 g) were ultrasonically extracted for 30 min in 15 mL n-hexane. The extracts were centrifuged at 4000 rpm for 15 min, and then the supernatant was recovered and concentrated by a rotary evaporator at 35 °C. For the clean-up, a column was prepared using 2 g of activated silica gel and 2 g of anhydrous sodium sulfate (Merck, Rome, Italy) in a glass column plugged with glass wool and eluted with n-hexane. The eluate was collected in a flask, adjusted to 5 mL with n-hexane and analyzed via GC-MS (8860 GC System with a single quadrupole mass spectrometer detector5977 MSD, Agilent, Santa Clara, CA, USA) equipped with a HP-5ms column (19091S-433, Agilent). The GC oven temperature method was as follows: initial temperature at 60 °C, hold for 3 min, then ramp at 15 °C/min to 320 °C. The mass selective detector operated in scan mode. Helium was used as the carrier gas at the flow rate of 1 mL/min. The concentrations of total hydrocarbons (total petroleum hydrocarbons, TPH_C10–C28_) and of the n-alkanes in the range n-decane (C10)–n-tetracosane (C40) were obtained via interpolation with calibration curves using the standard homologous series of n-alkanes (Linear Hydrocarbon Mixture C10–C40 UST-400-1, Ultra Scientific Italia, Bologna, Italy) and diesel fuel at different concentrations. Three independent samples for each experimental condition were analyzed.

### 2.3. Statistical Analysis

Data were statistically analyzed by Past4.04 program ANOVA, and Tukey–Kramer tests were applied for the comparison of multiple samples, since normality and homogeneity of variance were satisfied for all datasets. In particular, the ANOVA test was used to check the significance of differences in the dry weight of, alternatively, the vetch and of the rye (considering roots, shoots and the whole plants) in relation to the different diesel concentrations (1 factor, contamination, 4 levels, fixed; S0 vs. S1 vs. S2 vs. S3).

## 3. Results

### 3.1. Production of Plants Containing Diesel-Derived Hydrocarbons: Survival and Growth

The plant biomass necessary for setting up the analytical methodology to assess the content of diesel-derived compounds in plant tissues was obtained by growing two plant species, belonging to two different botanical families, on a soil spiked with increasing concentrations of diesel fuel.

Independently of contaminant concentration, nearly all the seeds sown (65 per pot) germinated, producing plantlets that grew until the end of the experiment (30 days). Only a slight delay in germination of rye seeds was observed at the highest concentration used (S3), in agreement with ref. [21].

Figure 1 shows the mean biomass of plants (measured as mean dry weight) obtained in the different contaminated and uncontaminated soils after 30 days from germination. No statistical difference was observed among the soils (ANOVA, *p* < 0.05), although, with increasing diesel contamination, the growth of rye plants tended to decline, and at the maximum diesel concentration (S3), many plants exhibited chlorosis, likely due to the interference of diesel compounds with mineral uptake [22]. However, data suggest a great tolerance of both common vetch and rye to diesel hydrocarbons that allowed us to obtain the necessary amount of biomass for the setup of the analytical protocol.

### 3.2. Measurements of Diesel-Derived Hydrocarbons in Soil and Plant Tissues

By applying the analytical method described in the Materials and Methods section, we analyzed all the soils and plant biomasses that we considered in our experimental design.

Figure 2 shows two chromatograms (Figure 2A,B) and their overlap (Figure 2C); the chromatogram in Figure 2A is representative of a chromatographic analysis of diesel, while the one in Figure 2B is representative of what was obtained by analyzing a representative sample of the above-ground biomass of common vetch grown in unpolluted soil and thus showing the biogenic hydrocarbons of common vetch shoot. In Figure 2A, it can be observed that most diesel hydrocarbons are in the n-alkanes range C10–C28 (retention time (RT) < 19.5 min), as well as the typical unresolved complex mixture signal (UCM). On the other hand, in Figure 2B, it can be noticed that the plant linear hydrocarbons, even if present, are negligible in this range, while the areas of the more abundant biogenic alkanes are beyond C28 (RT > 19.5 min). This is the same for the shoot and it is even more evident for roots of all the other species analyzed in this and other works in the literature [23]. Thus, in the specific case of diesel, it is possible to use the parameter TPH_C10–C28_ or for greater caution TPH_C10–C26_ to distinguish between diesel-derived and biogenic hydrocarbons.

Figure 3 shows the overlapping of representative chromatograms related to root (A) and shoot (B) extracts from common vetch plants grown in contaminated (S3, brown chromatograms) and uncontaminated (S0, green chromatograms) soil for 30 days. It can be observed that plants grown in contaminated soil took up diesel compounds, accumulating most of them in root tissues.

The concentration of diesel-derived hydrocarbons both in soil and plant organs, measured at the end of the exposure (T30), increased in parallel to the increase in soil diesel concentration (Figure 4). The correlation between diesel HC concentration in soil and plant organs was very high, r^2^ ranging from 0.92 to 0.99, indicating the reliability of the methodology. A similar correlation was observed for pots without plants (Figure 5) where soil was analyzed at the beginning (T0) and at the end of the experiment (T30). However, although the concentrations of diesel HCs determined in all spiked soils at T0 were proportional to the amount of spiked diesel fuel, they were much lower than expected (from 25 to 50%), likely due to the loss of volatile compounds present in diesel fuel [24]. At T30, diesel fuel concentration in soils without plants was statistically lower that at T0 but higher than that determined in soils with plants, suggesting that both natural attenuation and plants contributed to the decrement of soil hydrocarbon concentration.

The GC-MS-based methodology was also applied to analyze common cattail plants grown in polluted soil from a paddy field highly contaminated by diesel and in unpolluted soil from the same field for comparison. Figure 6 shows that petrogenic hydrocarbons were detected only in plants grown on highly contaminated soil and accumulated mainly in the roots. In the same contaminated soil, common vetch and rye did not produce enough biomass to be analyzed. The species common cattail was selected for its high tolerance to hydrocarbons.

## 4. Discussion

Plants are primary producers, the gateway for energy to enter the biosphere. As the largest component of the Earth’s biomass at the base of all food webs, plants are also a means of transfer to other organisms for the toxic substances they can absorb from the environment, which accumulate in their tissues through their metabolism. Quantifying the transfer of contaminants from the environment into terrestrial plants is then essential for assessing human and ecological risks, predicting phytoremediation effectiveness and biomass disposal [25].

However, while the methodology used to quantify the uptake of inorganic chemicals in plants is well established, effective standardized protocols to disentangle biogenic from petrogenic hydrocarbons in plants growing in polluted soils are not yet available.

Recently, Hunt et al. [12] have critically reviewed the available literature about the plant uptake and accumulation of petroleum hydrocarbons (PHC). They obtained variable results with respect to the extent of PHC accumulation in plant tissues, and one of the main causes was indeed the lack of a standardized methodology in detecting PHC in plants.

The same issue was encountered by Doucette et al. [25] in their analysis of land plant bioaccumulation data for different classes of organic chemicals. They showed the lack of comparable high-confidence data, which limited model evaluation and development.

In this work, we demonstrated that the set-up GC-MS-based protocol described in the Materials and Methods section and graphically shown in Figure 7 was able to assess the concentration of diesel-derived compounds in our two model land plants, common vetch and rye, distinguishing them from those of biogenic origin.

This protocol is particularly important for the management and remediation of diesel-contaminated agricultural soils as diesel is thought to be more toxic to organisms than crude oil because of its high light hydrocarbon content.

The recent Italian regulation for soil recovery in agriculture and husbandry areas (DM 2019, n.46) suggests phytoremediation as clean up technology and, in keeping with other environmental agencies (i.e., U.S. EPA), considers the ingestion of plants exposed to contaminants a pathway of exposure for humans. It follows that in this applicative sector, a reliable measurement of the level of hydrocarbons in crops for risk assessment and in remediating plants is particularly important. Indeed, during diesel phytoremediation, plants are growing on contaminated soil, promoting hydrocarbon degradation. Usually, most degrading activities occur in the rhizosphere thanks to the root-associated microorganisms [26,27]. However, plants can also uptake hydrocarbons, degrading and/or accumulating them in root and/or shoot [12]. Thus, these plants can potentially contain diesel-derived hydrocarbons that should be quantified to understand the process and to manage biomass disposal. From a circular economy perspective, this is also essential to valorise the material, avoiding any risk related to contaminants.

Until now, several methods have been used to extract and quantify diesel hydrocarbons, all developed for soil analysis and not for plants. According to the most currently applied methods, ISO 16703:2004 [15] and the Canadian Council of the Ministers of the Environment (CCME) method, the analysis of soil hydrocarbons is based on their extraction with the combination of a polar organic solvent and a non-polar one [28,29], followed by their quantification via gas chromatography with a flame ionization detector (GC-FID).

Both methods are not suitable to assess the concentration of diesel-derived hydrocarbons in plant tissues because they are unable to distinguish diesel-derived compounds from plant natural hydrocarbons. Specifically, the solvents extract both petrogenic and biogenic compounds whose retention times overlap in GC-FID analysis [30,31]. Although the standard analytical methodologies indicate a purification step to remove the plant-derived polar compounds from the extract before the GC-FID analysis, a few non-polar biogenic hydrocarbons remain in the sample, affecting the determination. For this reason, we set up a protocol based on GC-MS instrumentation. After demonstrating that hexane is the best solvent for extracting petrogenic compounds from the plant organs, and having removed any remaining polar biogenic compounds from the sample through an activated silica column that binds their polar functional groups, we analyzed the chromatograms obtained via GC-MS. In agreement with Wang et al. [30], our chromatograms, obtained from plants grown in uncontaminated soils, showed the absence of the unresolved complex mixture of hydrocarbons (UCM) and are predominated by resolved peaks related to n-alkanes in the high molecular carbon C27–C31 range. The absence of UCM suggests that nearly all polar biogenic compounds were effectively removed by the silica gel cleanup and that the n-alkanes detected in these samples are typical of terrestrial plants. Indeed, many studies showed that terrestrial plants typically synthesize long chain n-alkanes as part of the epicuticular leaf wax, contributing to their hydrophobic properties and protecting the leaf from the external environment [32]. A great deal of research has been devoted to identifying, quantifying, and interpreting naturally occurring leaf wax n-alkanes in plants, often with the goal of using them as taxon-specific chemical fingerprints. With this aim, Bush and McInerney [23] determined the abundance variation of n-alkanes within trees and across plant functional types, combining the analysis of some terrestrial plants with a meta-analysis of the published n-alkane literature (totalling 2093 n-alkane measurements from 86 sources). They confirmed that terrestrial plants (non-vascular and vascular species) generally produce n-alkanes in the molecular carbon C21–C37 range, commonly with a strong odd-over-even predominance and one or two dominant chain lengths [33,34]. The dominant chain lengths of grass and woody Angiosperms were shown to be C29 and C31, the same we obtained for rye, although we found a relatively consistent amount of C27. Moreover, the authors demonstrated that in terrestrial vascular plants, the amounts of n-alkanes in the carbon C21–C25 range is largely absent; these chain lengths are instead predominantly found in Sphagnum mosses. Accordingly, we also found n-alkane chain lengths > C27 in leaves of common vetch and in the above-ground biomass of common cattail with the predominance of C31–C33 and C29, respectively. In the roots of our model plants, we instead detected negligible amounts of non-polar n-alkanes.

Based on these observations, and given the GC-MS distribution of diesel-derived hydrocarbons, mainly in the C11–C24 range, in crops and phytoremediation vascular plants, it seems possible to discriminate diesel-derived hydrocarbons from biogenic compounds and to determine their concentration by measuring the total hydrocarbons in the C10–C26 range. Accordingly, under our experimental conditions, all petrogenic compounds absorbed by the model plants were in the range of C10–C26. Among the alkanes, those most accumulated in the roots were C21 and C25, although their relative abundance depended on the level of contamination. Specifically, in common vetch roots, the most abundant were C25 and C21, respectively, in the lowest (S1) and highest (S2 and S3) contamination levels. In rye roots, C25 was always the most abundant, although the concentration of C21 increased at higher contamination levels. In shoots of both species, the most abundant petrogenic alkane was C25, although substantial amounts of C14, C16, C18 and C20 were translocated from roots to shoots.

Anyway, previous studies have suggested that n-alkane chain-length distributions may be influenced by the environment, possibly in addition to genetic controls. Indeed, variation in the abundances of long n-alkane chain lengths may be responding, in part, to local environmental conditions. Thus, although a recent work [23] suggests that these changes are limited and are related to n-alkane heavy chains (>C27), in the GC-MS analysis of diesel hydrocarbons in crops or phytoremediation plants, it should be good practice to insert a sample of the species grown in the same environmental conditions to confirm the absence or negligible amount of biogenic compounds in the n-alkane C11–C26 range.

## 5. Conclusions

Overall, this work shows a GC-MS-based protocol suitable for reliably determining the number of diesel-derived hydrocarbons accumulated by plants growing in contaminated soil. This methodology is specific to diesel contamination and cannot be applied to quantify all types of petrogenic contaminants (e.g., crude oil) in plant tissue, given the presence in plants of naturally synthesised hydrocarbons whose retention times overlap with those of petrogenic n-alkane heavy chains (>C28). In the specific case of diesel, we have shown that the use of a solvent that favors the extraction of non-polar compounds together with a purification step to eliminate polar biogenic compounds, followed by GC-MS analysis, offers the possibility to discriminate biogenic from diesel-derived compounds, allowing for the determination of these soil contaminants in plants. This protocol is particularly important in the field of soil phytoremediation. It will allow researchers to better understand the mechanisms behind the uptake, accumulation and degradation of diesel hydrocarbons in plants and enable technicians to perform reliable risk assessment during the phytoremediation of agricultural soils. The protocol will also allow operators to define how to dispose of the biomass produced by the phytoremediation of diesel-contaminated environments from a circular economy perspective.

## Figures and Tables

**Figure 1 plants-13-00298-f001:**
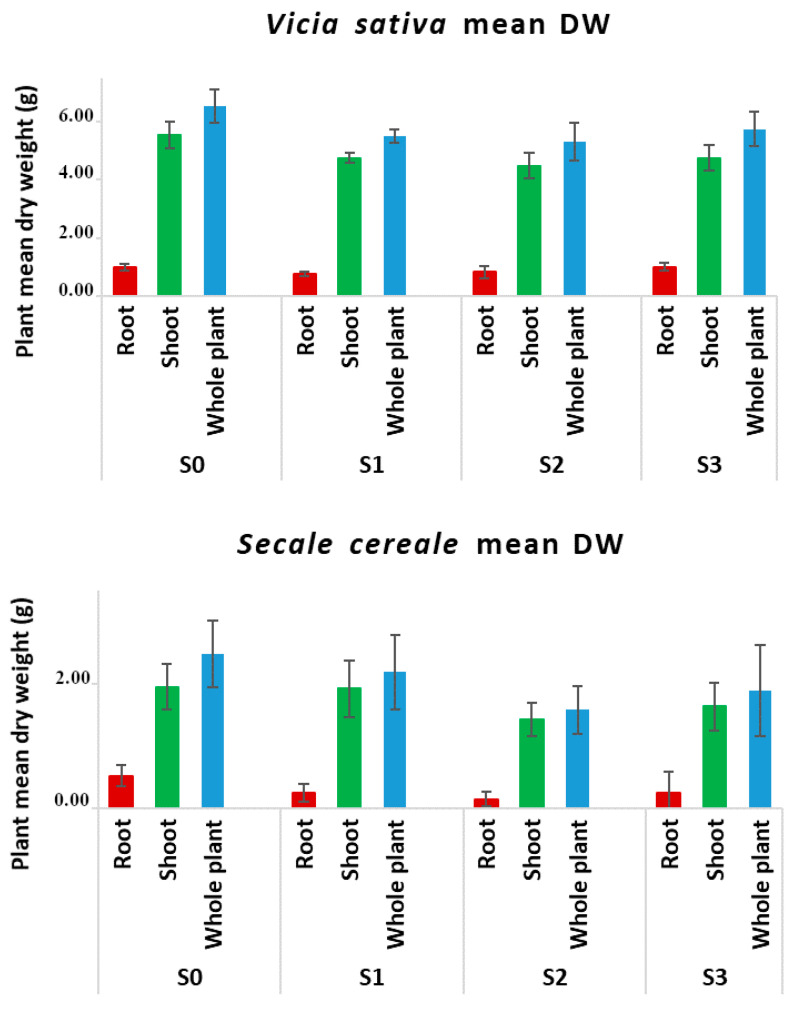
Plant dry biomass (mean DW ± SD) related to diesel fuel concentration in soil. The DW of the organs of plants grown in three independent pots for each experimental condition was determined, and the mean DW ± SD of the three repetitions is reported. The development of both common vetch and rye was not statistically affected by any tested diesel concentration (ANOVA, *p* < 0.05).

**Figure 2 plants-13-00298-f002:**
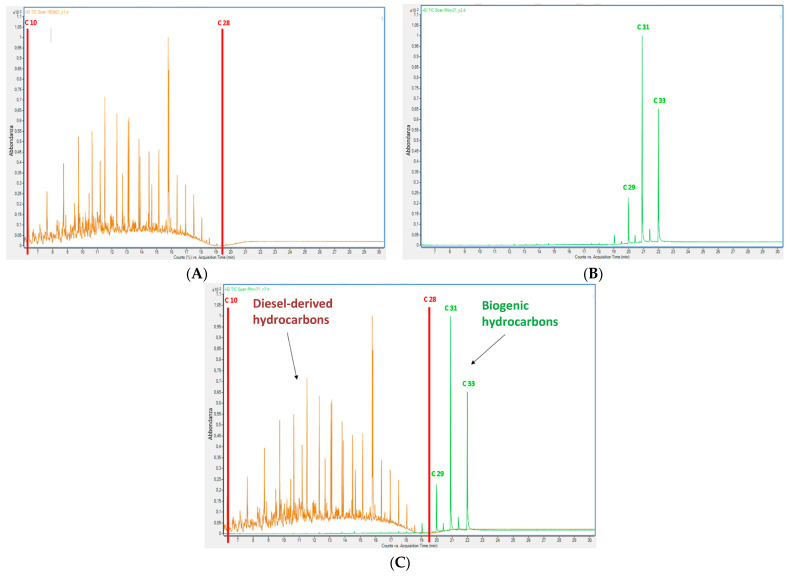
Representative chromatograms of diesel-derived hydrocarbons (**A**) and biogenic hydrocarbons (**B**) obtained via GC-MS analysis of a diesel fuel sample and common vetch shoot biomass, respectively. Panel (**C**) shows the overlap of the two profiles. The areas of the more abundant biogenic alkanes (green profile) are beyond C28 (RT > 19.5 min), whereas they are negligible in the n-alkanes range C10–C28 (RT < 19.5 min) typical of diesel-derived hydrocarbons (brown profile).

**Figure 3 plants-13-00298-f003:**
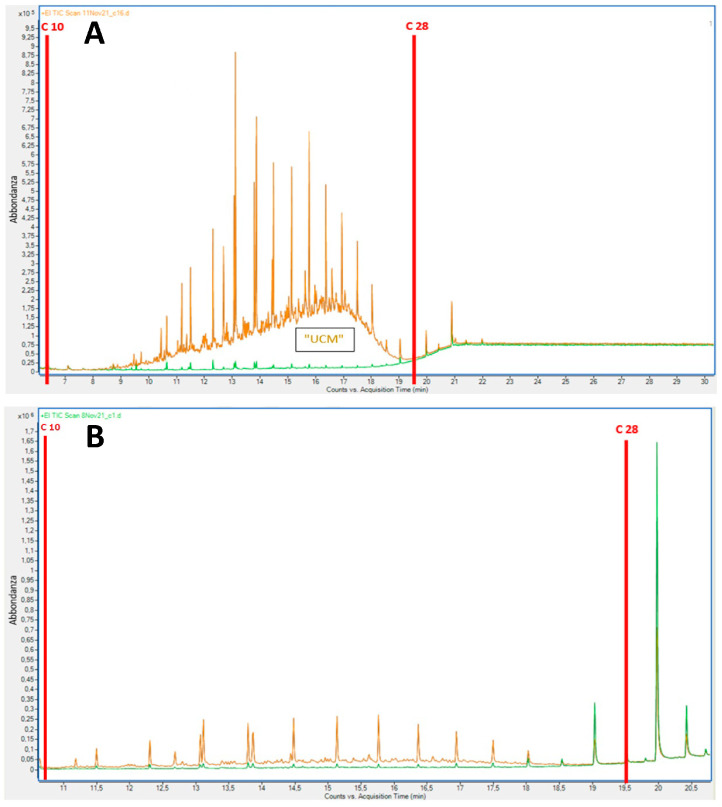
Overlap of representative GC-MS chromatograms of root (**A**) and shoot (**B**) extracts from common vetch plants grown in contaminated (S3, brown chromatograms) and uncontaminated (S0, green chromatograms) soil for 30 days. Plants grown in contaminated soil accumulated most diesel-derived hydrocarbons in root tissues.

**Figure 4 plants-13-00298-f004:**
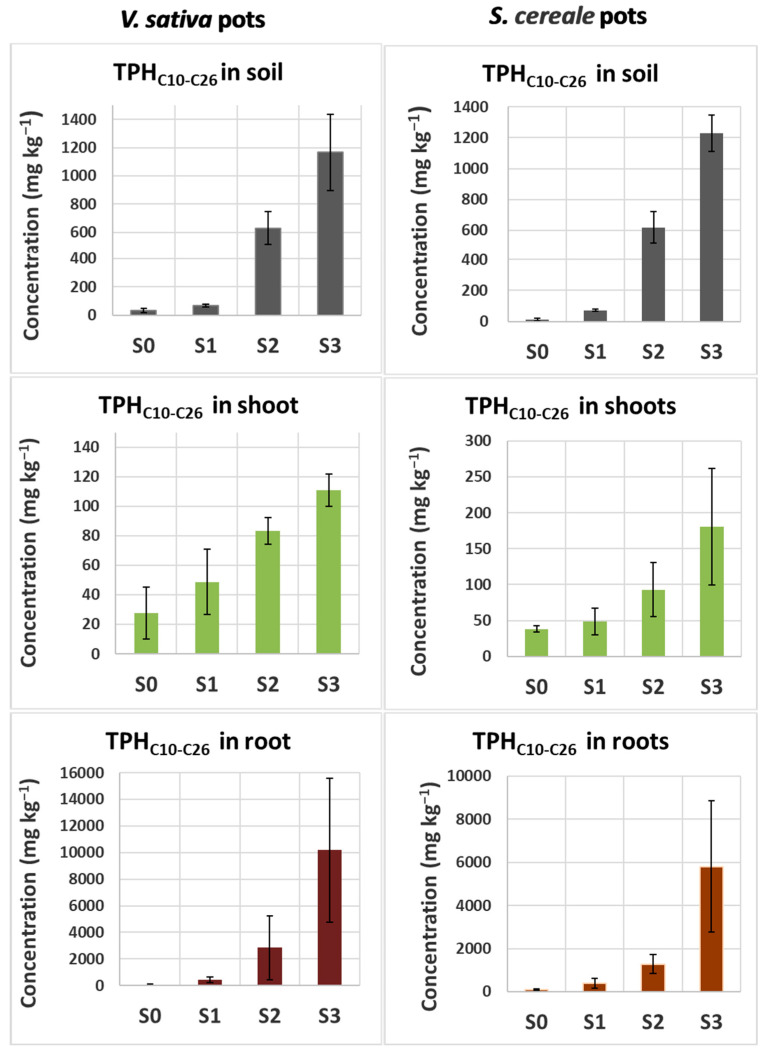
Total diesel-derived hydrocarbons detected in the root (brown) and shoot (green) of common vetch and rye plants grown in soils with increasing concentration of diesel fuel. Total diesel-derived hydrocarbons were calculated as TPHs in the range C10–C26. The species accumulated diesel compounds in both their roots and shoots with a concentration positively correlated with that in soil. However, most of them were accumulated in roots. Three independent samples for each experimental condition were analyzed, and the reported concentrations are the mean ± SD.

**Figure 5 plants-13-00298-f005:**
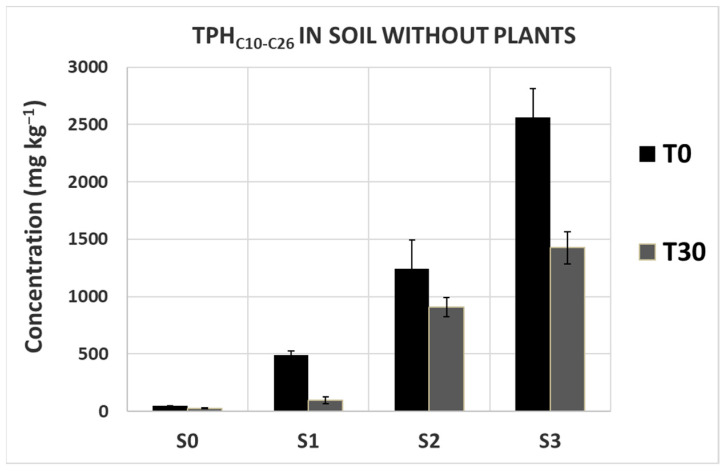
Total diesel-derived hydrocarbons detected in soils spiked with diesel fuel at the beginning (T0, black) and at the end of the experiment (T30, grey). The concentration was calculated as TPHs in the range C10–C26. The reported concentrations are the mean ± SD of three measurements on three independent samples for each experimental condition.

**Figure 6 plants-13-00298-f006:**
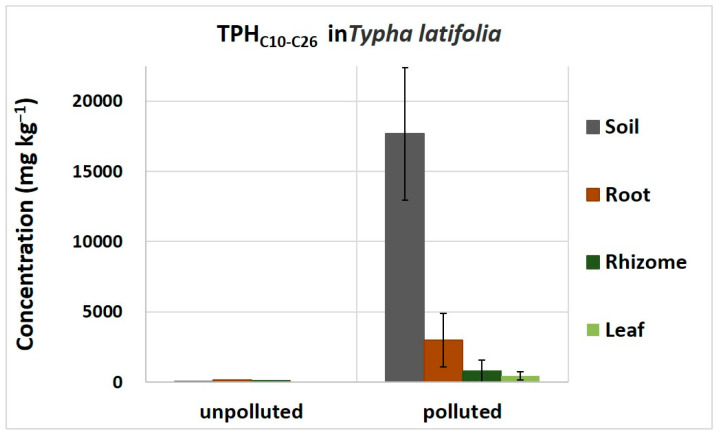
Total diesel derived hydrocarbons accumulated in common cattail organs after 30 days of exposure. Diesel compounds were mainly accumulated in the root. The reported concentrations are the mean ± SD of three measurements on three independent samples from polluted and unpolluted condition.

**Figure 7 plants-13-00298-f007:**
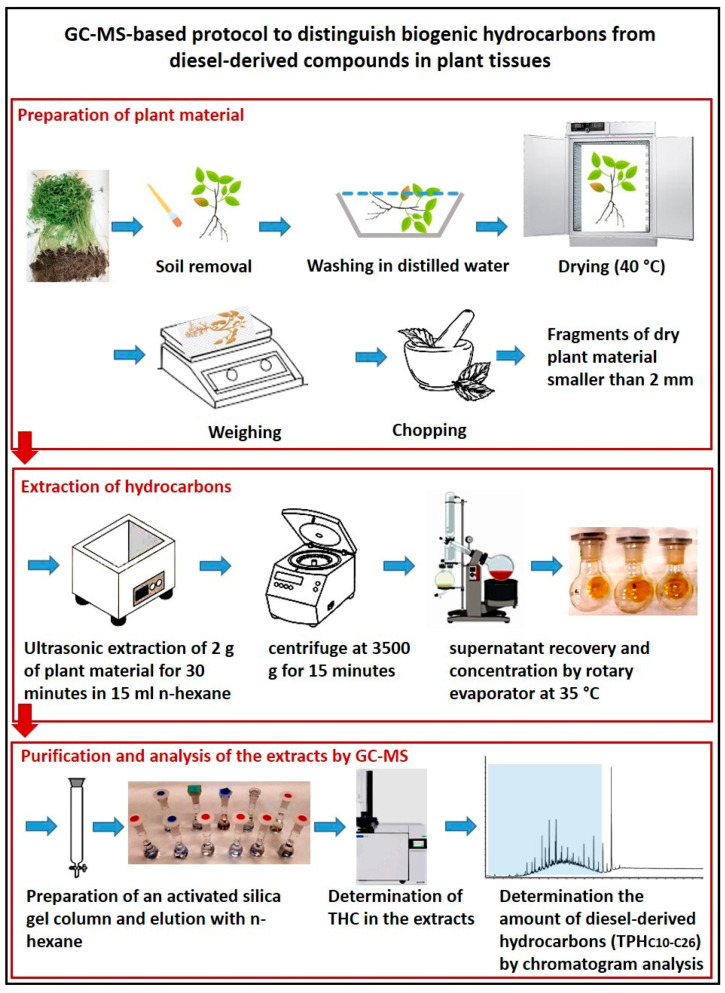
Graphic representation of the GC-MS-based protocol for the determination in plant tissues of compounds deriving from diesel as a soil contaminant.

**Table 1 plants-13-00298-t001:** Experimental conditions and related number of pots set up for each condition.

	S0	S1	S2	S3
	(0 mg kg^−1^)	(1000 mg kg^−1^)	(5000 mg kg^−1^)	(10,000 mg kg^−1^)
No plants	3 pots	3 pots	3 pots	3 pots
*Vicia sativa*	3 pots	3 pots	3 pots	3 pots
*Secale cereale*	3 pots	3 pots	3 pots	3 pots

## Data Availability

Data are contained within the article.

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
