# Peer review of "Analysis of Petrogenic Hydrocarbons in Plant Tissues: A Simple GC-MS-Based Protocol to Distinguish Biogenic Hydrocarbons from Diesel-Derived Compounds"

_plants, 2024, doi:10.3390/plants13020298_

Round 1
Reviewer 1 Report
Comments and Suggestions for Authors
The manuscript entitled " Phytoremediation of diesel-contaminated farming soils: a GC-MS-based protocol to distinguish biogenic hydrocarbons from diesel-derived compounds in plant tissues", introduced a protocol to enable the determination of diesel derived hydrocarbons in contaminated plants. After carefully reviewing the manuscript authored by Collina et al., it is evident that the authors conducted a meticulous study to evaluate a GC-MS based protocol in identify diesel derived hydrocarbons using Vicia sativa L. and Secale cereale as phytoremediation plant models. The article is commendably written, and the concepts are presented with great clarity. The manuscript offers valuable insights into the ways we can detect hydrocarbons from plant species from contaminated soils. However, I have some minor comments that authors should address before publication, listed below:
1: Authors should indicate the error bars in Figures 1, 4, 5 and 6 to indicate the statistical reliability of the results attained in this study since ANOVA was used for statistical analysis thereby indicating the experiments were conducted in triplicates
2: A graphical schematic representation denoted as a new Figure 1 could be added to depict the experimental flow and layout.
3 Among the alkanes group that was mostly absorbed by the plant species used, which of them were mostly absorbed by the model plants from the contaminated soils?
4: Page 9, line 298, what is the meaning of the letter 'e'?
5: How and why is the concentration in line 332 different from the other 4 stated in Table 1?
Author Response
1: Authors should indicate the error bars in Figures 1, 4, 5 and 6 to indicate the statistical reliability of the results attained in this study since ANOVA was used for statistical analysis thereby indicating the experiments were conducted in triplicates.
The meaning of the bars inserted in the Figures has been explained in the legend of each figure.
2: A graphical schematic representation denoted as a new Figure 1 could be added to depict the experimental flow and layout.
In order to combine your suggestion with that of the reviewer 3, we have moved the Material and Methods before the results and we have prepared and added a new figure graphically showing the experimental protocol.
3 Among the alkanes group that was mostly absorbed by the plant species used, which of them were mostly absorbed by the model plants from the contaminated soils?
The most absorbed alkanes in the roots of the model plants (Vicia sativa and Secale cereale) were C21 and C25, but their relative abundance in the roots depended on the level of contamination.
In the roots of Vicia sativa the most abundant were C25 and C21 at the lower (1000 mg kg-1) and higher (5000 or 10000 mg kg-1) contamination respectively.
In the roots of Secale cereale the most abundant was always C25, even if the concentration of C21 increased at higher contaminations.
In shoots of both species the most abundant petrogenic alkane was C25, although substantial amounts of petrogenic alkanes C14, C16, C18 and C20 were also translocated from the roots.
This information has been added to the Discussion
4: Page 9, line 298, what is the meaning of the letter 'e'?
Sorry it was a mistake “e” has been replaced with “and”
5: How and why is the concentration in line 332 different from the other 4 stated in Table 1?
We set up the protocol for the determination of the amount of diesel derived hydrocarbons in controlled experimental conditions by growing the model plants Vicia sativa and Secale cereale in soils artificially contaminated (we decided the diesel concentrations). After setting up the protocol, we further evaluated its sensitivity and reliability by analysing an additional species, highly tolerant to contaminants and belonging to a different botanical family, the Typha latifolia grown on a soil from a diesel-contaminated site. When we determined the diesel concentration in this soil (a paddy field), we found a concentration higher than those we used for the setting up of the protocol. We tried to grow Secale cereale and Vicia sativa on this type of soil but we do not obtained the quantity of biomass needed to carry out GC-MS. We have better explained this part in the manuscript.

Reviewer 2 Report
Comments and Suggestions for Authors
Dear Authors,
I see the potential merit including novelity of yours work. However there are a lot of weaknes mainly in the way you are writting.
The title doesn’t match to the content of the manuscript. Soil used in pot experiment isn’t farming. It sand and undescribed organic matter.
The soil, where Typha latifolia was growing is also not described. ‘Natural soil’ and 'contaminated' is not a sufficient description.
Remove 'farming' from the tittle. Consider to replace 'Phytoremediation' on bio or phyto accumulation. There is no 'Phytoremediation' in Materials and Methods and Results chapters. It is also used in the Discussion chapter however never with self-results always as discussion.
Please be much more carefull in description of yours experiment.
Match the aim of the study to the Results chapter. The aim is dedicated to ‘set up a (…) protocol’ appropriate to disentangle the petrogenic content from the biogenic one in plant tissues (…) they are exposed to diesel derived hydrocarbon contaminated soil’. The results are about contents of hydrocarbons in chosen plants. The text following the aim is not clear. Is the part of the Authors’ aim or state of art? ‘This would allow a better understanding of the mechanisms of diesel hydrocarbon uptake, accumulation and degradation in plants and a reliable risk assessment during cropland phytoremediation. The protocol was also developed with the aim of defining how to dispose of the biomass produced by phytoremediation of diesel-contaminated environments in a circular economy perspective’. In the manuscript there are no methods, results and discussion for them.
There is no independent Conclusion chapter. The short paragraph in discussion chapter is not enough.
The materials and methods chapter is after results. I makes reading and understanding the manuscript more difficult. Please change the manuscript/chapters order.
Please be wary using following words: ‘uptake’, 'transfer and translocation’ and ‘tissues’. The cytological view would be valuable here. Did you consider a hypotesis of apoplast (cell wall system) 'uptake' and transport from roots to shoots without entering the cell (i.e. crossing cell membrane and entering cytoplasm)?
Self-results are not discussed in the discussion chapter. The differences in three plants researched are not compered. There are no hypothesis, why is that.
My detailed questions are as following:
1. Why did you sterilize the sand? And organic matter? Apart random microbes colonisation during substrate stabilisation (two weeks line 310), apart rhizobial strains (common vetch cultivation) what was the source of microbes? Was the seed treated with any pesticides before sowing? When you wash the common vetch roots were there the nodules on the roots? Were 30day in conditions of yours experiment enough to let Rhizobium sp. colonised the common vetch roots?
2. Why didn’t you use silt and clay? Why didn’t you use a soil taken from a normal for your University place farmland soil?
3. Please consider to write with in the manuscript: common vetch and rye and in the Material and method chapter write full latin names. I am aginst writting such abbverations like: V. sativa or S. cereale.
4. Please consider using ‘g’ instead of ‘mg’ for instance (table 1.) S1 1g*kg-1 instead of 1000 mg *kg-1
5. Why did you decide to terminate the pot experiment after 30days? What was the reasons?
6. Please carefully describe in the manuscript the method of removing the roots from the soil and cleaning? Washing? Separating the roots and the soil. Is there any chance, that root hairs did not stay with the soil?
7. What was the reasons for the light, temperature, day and night duration you took? How does it suite to normal cultivation conditions in yours region? What was the date of seed sowing? Was it sowing time for those two plants?
8. Claryfy Typha latifolia was also growing in pots? How was it planted? It is a water plant, so what was the soil taken for the experiment? What was the uncontaminated soil as control for Typha plant? Why did you decide to join this plant to the rye and common vetch?
9. Why do you use in one study parametric and non-parametric statistical tests?
Yours sincerely
Author Response
1: The title doesn’t match to the content of the manuscript. Soil used in pot experiment isn’t farming. It sand and undescribed organic matter. Remove 'farming' from the tittle. Consider to replace 'Phytoremediation' on bio or phyto accumulation. There is no 'Phytoremediation' in Materials and Methods and Results chapters. It is also used in the Discussion chapter however never with self-results always as discussion.
The title has been changed: the terms farming soil and phytoremediation have been removed from the title.
2: The soil, where Typha latifolia was growing is also not described. ‘Natural soil’ and 'contaminated' is not a sufficient description. Please be much more carefull in description of yours experiment.
The section regarding Typha latifolia has been improved and in particular the growing soil was carefully described in Material and Methods.
3: Match the aim of the study to the Results chapter. The aim is dedicated to ‘set up a (…) protocol’ appropriate to disentangle the petrogenic content from the biogenic one in plant tissues (…) they are exposed to diesel derived hydrocarbon contaminated soil’. The results are about contents of hydrocarbons in chosen plants. The text following the aim is not clear. Is the part of the Authors’ aim or state of art? ‘This would allow a better understanding of the mechanisms of diesel hydrocarbon uptake, accumulation and degradation in plants and a reliable risk assessment during cropland phytoremediation. The protocol was also developed with the aim of defining how to dispose of the biomass produced by phytoremediation of diesel-contaminated environments in a circular economy perspective’. In the manuscript there are no methods, results and discussion for them.
The aim of the work has been better stated in order to be consistent with the results. The sentence after the aim of the manuscript intended to underline the importance of the aim (the importance of setting up such methodology) but we are now aware that this is confusing to the reader. We moved and modify it.
4:There is no independent Conclusion chapter. The short paragraph in discussion chapter is not enough.
An independent “Conclusion” section has been introduced.
5: The materials and methods chapter is after results. I makes reading and understanding the manuscript more difficult. Please change the manuscript/chapters order.
Done
6: Please be wary using following words: ‘uptake’, 'transfer and translocation’ and ‘tissues’. The cytological view would be valuable here. Did you consider a hypotesis of apoplast (cell wall system) 'uptake' and transport from roots to shoots without entering the cell (i.e. crossing cell membrane and entering cytoplasm)?
Yes, but this is outside the scope of this work. Using the protocol developed, it is only possible to determine the amount of petrogenic compounds in plant organs independently of their distribution in cellular compartments and the mode of transport.
7: Self-results are not discussed in the discussion chapter. The differences in the plants researched are not compered. There are no hypothesis, why is that.
The difference between our GC-MS based protocol and the GC-FID based methodology currently applied to determine the concentration of petrogenic hydrocarbon in plants grown on contaminated soils was already discussed. We have improved the discussion e we added a few sentences on the differences in petrogenic compounds absorbed and accumulated by the model plants.
Detailed questions are as following:
- Why did you sterilize the sand? And organic matter? Apart random microbes colonisation during substrate stabilisation (two weeks line 310), apart rhizobial strains (common vetch cultivation) what was the source of microbes? Was the seed treated with any pesticides before sowing? When you wash the common vetch roots were there the nodules on the roots? Were 30day in conditions of yours experiment enough to let Rhizobium sp. colonised the common vetch roots?
The quartz sand and the potting compost were autoclaved to reduce the microbial load because the experiment was carry out in a plant growth room and it is a good practice in order to reduce the presence of plant pathogens. It also a standard procedure to homogenize the substrate. In our experience (detection and study of plant-microrganism interaction mainly through NGS sequencing and fluorescence microscopy) after the sterilization by autoclave (20 min at 121 °C) most microorganisms remain alive. We have added to the M&M the autoclave conditions. The seeds were no treated with any pesticide; common vetch seeds were inoculated with the specific rhizobia strain by mixing them with the ALOSCA granules that are dry, clay-based granules containing the rhizobia (https://www.alosca.com.au/). We have added the procedure to the M&M. As expected (usually in our conditions nodules can be seen with the naked eye in 2 weeks), after 30 days we noticed nodules on roots. We added this information to the text.
- Why didn’t you use silt and clay? Why didn’t you use a soil taken from a normal for your University place farmland soil?
We did not use silt and clay or agricultural soil to facilitate collection and cleaning of the roots and to be sure to collect most of the root material and that the measured petrogenic compounds came from the roots and not from small grains of soil attached to the roots. We have added a sentence in M&M.
- Please consider to write with in the manuscript: common vetch and rye and in the Material and method chapter write full latin names. I am aginst writting such abbverations like: V. sativaor S. cereale.
Done
- Please consider using ‘g’ instead of ‘mg’ for instance (table 1.) S1 1g*kg-1instead of 1000 mg *kg-1
Thanks for the suggestion, but we prefer to keep mg kg-1 because it corresponds to ppm, which is the unit usually used by technicians in environmental monitoring and remediation.
- Why did you decide to terminate the pot experiment after 30days? What was the reasons?
The duration of 30 days resulted in healthy plants that were likely to have ad/absorbed diesel-derived compounds. A longer growing time under our experimental conditions would have affected the health of the plants. We added a sentence in M&M.
- Please carefully describe in the manuscript the method of removing the roots from the soil and cleaning? Washing? Separating the roots and the soil. Is there any chance, that root hairs did not stay with the soil?
We have described more carefully the root collection in M&M. About root hairs, and fine root in general it is likely that a negligible part remained in the soil, but in our experience it is not possible to recover all the roots.
- What was the reasons for the light, temperature, day and night duration you took? How does it suite to normal cultivation conditions in yours region? What was the date of seed sowing? Was it sowing time for those two plants?
The experimental conditions in terms of temperature and light (day and night duration) were suitable for growing common wetch and rye independently from their period of seed sown in field in our region. Our objective was only to obtain healthy plants in artificial standard conditions.
- Claryfy Typha latifoliawas also growing in pots? How was it planted? It is a water plant, so what was the soil taken for the experiment? What was the uncontaminated soil as control for Typha plant? Why did you decide to join this plant to the rye and common vetch?
The section regarding Typha latifolia has been improved. All the information have been reported at the end of the Introduction and in Material and Methods
- Why do you use in one study parametric and non-parametric statistical tests?
The distribution of different datasets (dry biomass, plant height, hydrocarbon content, etc.) is not always normal and in these cases it is necessary to transform the data or apply non-parametric statistical tests. In the first version of our paper, we had included data on some plant traits that were not normally distributed, which is why we claimed to use both parametric and nonparametric statistical tests. However, we then decided to remove them because they were redundant. Therefore, now the document contains only normally distributed datasets that have been analyzed by ANOVA. Accordingly, the paragraph relating to statistics has been modified, sorry for this typo.

Reviewer 3 Report
Comments and Suggestions for Authors
It looks fine

Comments on the Quality of English LanguageIt looks fine
Author Response
1. In the objective parts, the author has to be more specific. Currently, there is no any risk based analysis in the manuscript as has been discussed by saying it would give the idea about it. Moreover, there is no any discussions comparing GC-MS based analysis and other methodologies, however have been discussed in the introduction section. Please make objective based on your findings only without comparing others. The objective of the work has been better stated in order to be consistent with the results. 2. Currently, some of the lines author wrote in result section would have been appropriate in material and method sections e.g. line 89-92, 114-115, 180-182 etc. author need to be very consistent. Material and Methods have been improved and moved before the Results 3. I would like to suggest to make schematic figure for the protocol based analysis at the findings (results/discussions) sections to clearly portray your idea to the common readers. A new Figure 7 graphically showing the protocol has been prepared and added to the Discussion.

Round 2
Reviewer 2 Report
Comments and Suggestions for Authors
Dear Editor,
I am glad with the work, the Authors has done. Please accept the manuscrpt.
yours sincerely Paweł J. Mundała